# Exploring the Potential of Anthocyanin-Based Edible Coatings in Confectionery—Temperature Stability, pH, and Biocapacity

**DOI:** 10.3390/foods13152450

**Published:** 2024-08-02

**Authors:** Carmo Serrano, Beatriz Lamas, M. Conceição Oliveira, Maria Paula Duarte

**Affiliations:** 1Instituto Nacional de Investigação Agrária e Veterinária (INIAV, I.P.), Av. da República, 2780-157 Oeiras, Portugal; 2Associated Laboratory TERRA, LEAF–Linking Landscape, Environment, Agriculture and Food–Research Center, Instituto Superior de Agronomia, University of Lisbon, Tapada da Ajuda, 1349-017 Lisboa, Portugal; 3The Mechanical Engineering and Resource Sustainability Center (MEtRICs), Chemistry Department, NOVA School of Science and Technology, Universidade NOVA de Lisboa, 2829-516 Caparica, Portugal; b.lamas@campus.fct.unl.pt (B.L.); mpcd@fct.unl.pt (M.P.D.); 4Centro de Química Estrutural, Institute of Molecular Sciences, Instituto Superior Técnico, Universidade de Lisboa, 1049-001 Lisboa, Portugal; conceicao.oliveira@tecnico.ulisboa.pt

**Keywords:** anthocyanins, edible coatings, confectionery, temperature stability, biocapacity

## Abstract

This study aims to develop purple-coloured polymeric coatings using natural anthocyanin and desoxyanthocianidins (3-DXA) colourants for application to chocolate almonds. The objective is to achieve a stable and uniform colour formulation throughout processing and storage, enhancing the appearance and durability of the almonds to appeal to health-conscious consumers and align with market demands. Plant materials like sweet potato pulp, sweet potato peel, radish peel, black carrot, and sorghum were employed to obtain the desired purple hue. Anthocyanidins and 3-DXA were extracted from the matrices using solvent extraction and ultrasound-assisted methods at different pH values. High-performance liquid chromatography with diode array detection (HPLC-DAD) and high-resolution tandem mass spectrometry (HRMS/MS) were used to identify the compounds in the extracts. The highest antioxidant capacities, as measured by the DPPH^•^ and FRAP methods, were observed in purple sweet potato and dye factory extracts, respectively; meanwhile, sorghum extract inhibited both α-amylase and α-glucosidase, indicating its potential for managing postprandial hyperglycemia and type 2 diabetes. The degradation kinetics of coloured coatings in sugar syrup formulations with anthocyanins and 3-DXA revealed that locust bean gum offered the best colour stabilization for plant extracts, with sorghum extracts showing the highest and black carrot extracts the lowest colour variation when coated with Arabic gum. Sweet potato pulp extracts exhibited less colour variation in sugar pastes, both with and without blue spirulina dye, compared to factory dye, highlighting their potential as a more stable and suitable alternative for colouring purple almonds, particularly over a five-month storage period. This study supports sustainable practices in the confectionery industry while aligning with consumer preferences for healthier and environmentally friendly products.

## 1. Introduction

The confectionery industry continually seeks innovative approaches to find colouring molecules that can compete with synthetic analogues, enhancing product appeal while meeting consumer demands for healthier options [1]. Another strategy in the food industry is to replace food additives by removing E codes from labels and using natural food ingredients with colouring properties. These strategies utilize less processed vegetable materials to achieve clean label solutions, maintaining food colour, quality, and safety. Due to these changes, the food industry aims to replace synthetic dyes such as Brilliant Blue (E133) and Allura Red (E129) with natural dyes that possess bioactive properties [2]. These natural dyes can be derived from leaf and peel co-products of the agri-food industry, transforming waste into valuable resources.

Anthocyanins are a group of water-soluble pigments responsible for the colours ranging from red to purple to blue in many fruits, vegetables, flowers, and leaves. The colours produced by anthocyanins have been studied for several factors, including light [3], temperature [4], pH [3,5], copigments [3,6,7,8], metal ions, and other environmental factors.

Some studies [3,6,7] have suggested that anthocyanin degradation is accelerated as temperature increases, and the loss of colour is more pronounced in the presence of phenolic acids and other flavonoids, such as rutin [3]. However, the stability of anthocyanins can be increased through copigmentation, which involves the association between anthocyanins and one or more flavonols through hydrogen bonds. This copigmentation forms a protective structure resistant to external factors such as pH. Other authors [3] have noted that in acidic conditions, anthocyanins tend to appear red, while in more neutral conditions, they can appear purple or blue [3]. The colour variation in anthocyanins with pH is due to the equilibrium of molecular species in solution as pointed out by [9].

The structures of anthocyanins can vary depending on the specific compound and the plant species from which they come. However, they typically consist of anthocyanidins derived from flavonoids. Anthocyanins are often glycosylated, meaning they have sugar molecules attached to the aglycone. These sugar groups can include glucose, galactose, arabinose, rhamnose, or other sugar molecules [3]. Some anthocyanins may also have acyl groups attached to the sugar molecules, such as acetyl, coumaroyl, caffeoyl, or organic acids (malonic, succinic, tartaric, and malic acid) [10]. Acylated anthocyanins are more stable to pH, temperature, and interaction with other matrix compounds than non-acylated anthocyanins [11].

Deoxyanthocyanidins (3-DXAs) are distinguished from anthocyanidins due to the absence of a hydroxyl group at position 3, an indication that at pH 1–7, no difference is observed in the colour of 3-DXAs extracted from sorghum [5]. However, some authors claim that it is possible to extract the violet colour from sorghum when using an aqueous solution of pH 9 [12]. In sorghum, the forms of 3-DXA are apigeninidin and luteolinidin, which give the yellow and red colour, respectively [13,14].

Several studies have been published on the extraction of these compounds using some solvents, such as alcohols [15], alkaline water [16], acidified water [15,16,17], or ionic liquids [18,19,20] to enhance the extraction efficiency. However, these methods are time-consuming, require larger volumes of solvents, are high-cost, and have potential environmental impacts. Other methods have been applied by other authors like ultrasound-assisted (UA) [21], microwave-assisted (MA) [22], and pulsed electric field (PEF) extraction [23], they are distinguished by increasing the availability of anthocyanins in the extraction solvent, thus increasing the extraction yield. However, they require specialized equipment and careful control of electrical parameters [16,24,25].

Apart from their role in plant pigmentation, anthocyanins have gained recognition in several studies for their antioxidant [18], anti-inflammatory, and potential anti-cancer properties. Furthermore, acylated and non-acylated anthocyanins have been reported for prebiotic, antidiabetic, and antioxidant properties, with the consumption of plant sources with these molecules playing an important role in health [11,26,27].

Edible coatings, commonly used in the food industry, are proteins, polysaccharides, or lipids. Generally, proteins are used to provide mechanical stability to the film, polysaccharides to control gas exchange, and lipids to reduce water transmission [28]. The choice of polymeric materials influences the texture of the films, giving shine or opacity, extending shelf life, enhancing sensory attributes, and improving nutritional value [29,30]. Integrating anthocyanin-rich extracts into edible coatings offers several properties by the formation of a uniform film to enhance the adhesion and stability of anthocyanins during the industrial process [28,31,32]. These coatings provide a protective barrier against moisture and light transmission, and act as an encapsulating agent, immobilizing anthocyanins and protecting them from light, heat, pH changes, and oxidation [3,29].

A crucial aspect of evaluating anthocyanin-based edible coatings is their temperature resistance. Arabic gum is one of the polymeric materials most used in coatings in the confectionery industry, due to its solubility in water and ability to form gels [33]. Locust gum, used in the food industry as a thickening, stabilizing and gelling agent, or emulsifier (E410), is more resistant to temperature than Arabic gum [34] and has antioxidant capacity [35]. This study aims to evaluate the resilience of these two coatings to temperature variations and their role in stabilizing anthocyanin-based dyes applied to sugar syrup, in terms of colour discolouration due to sugar syrup.

Another critical consideration in utilizing anthocyanin-based coatings is their sensitivity to pH values [31]. Obtaining a stable purple hue directly from anthocyanin- or 3-DXA-based plant matrices, without mixing blue and red dyes, would simplify the formulation process and offer more consistent colours throughout the product’s shelf-life.

Beyond their visual impact, the biocapacity of anthocyanins is a key attribute to their appeal in confectionery applications. Incorporating anthocyanin-rich coatings into confectionery products offers the potential to deliver antioxidant benefits to consumers. However, ensuring the retention of bioactive compounds throughout processing and storage poses a challenge.

This study intends to obtain a stable purple colour directly from anthocyanin-based plant matrices, specifically using leaves and peels from agri-food by-products, investigate the stabilization of anthocyanin-based dyes using Arabic and locust gum coatings, and evaluate the resilience of these coatings to temperature variations in terms of colour discolouration when applied to sugar syrup.

## 2. Materials and Methods

### 2.1. Chemicals and Materials

Ethanol p.a., citric acid, sodium citrate, gallic acid, hydrochloric acid, ascorbic acid, iron (II) sulphate heptahydrate, iron (III) chloride hexahydrate, and sodium acetate trihydrate were purchased from Merck (Darmstadt, Germany). Sodium carbonate anhydrous, sodium bicarbonate, Folin–Ciocalteu reagent, potassium hexacyanoferrate (III), and anhydrous sodium sulphate were purchased from Panreac (Barcelona, Spain). 2,4,6-tris (2-pyridyl)-S-triazine (TPTZ) and ferric chloride were acquired from Fluka (Buchs, Germany). Ethanol absolute anhydrous was purchased from Carlo Erba (Marseille, France). Other reagents such as potassium chloride, soluble potato, starch, sodium phosphate, 4-nitrophenyl-α-D-glucopyranoside, 6-hydroxy-2,5,7,8-tetramethylchroman-2-carboxylic acid (trolox), 2,2-diphenyl-1-picrylhydrazyl radical (DPPH^•^), 3,5-dinitrosalicylic acid (DNS), and the enzymes α-amylase from porcine pancreas and α-glucosidase from Saccharomyces cerevisiae were purchased from Sigma-Aldrich (Sternheim, Germany). Acarbose was purchased from Alfa Aesar (Karlsruhe, Germany). Food grade additives such as locust bean gum was purchased from Sosa (Ingredients S.L., Spain, and Arabic gum was purchased from Sigma-Aldrich (Darmstadt, Germany). The sugar (Sidul, Lisbon, Portugal) and the solid white sugar paste (Auchan) were purchased from a commercial store Auchan (Lisbon, Portugal). All other unlabelled chemicals and reagents were analytical or HPLC-MS Optima grade.

### 2.2. Plant Material

Fresh black carrots (*Daucus carota* ssp. sativus var. atrorubens), radishes (*Raphanus sativus*), and purple sweet potatoes (*Ipomoea batatas* (L.) Lam) were acquired from a commercial store, Celeiro, in Sintra, Portugal. Dried sorghum (*Sorghum bicolour* L.) was purchased from PMA28 in Varize, France, and ground in a mill (IKA Micro Fine Mill Culatti) using a 1.0 mm thick sieve. The powdered sorghum was stored under vacuum in a packaging film polymer (LDPE 60 µm/PA 30 µm) from Amcor Flexibles, Oeiras, Portugal, and placed in desiccators until further analysis.

### 2.3. Anthocyanins Extracts

Anthocyanins from black carrot, sweet potato (peel and pulp), and radish (peel) were extracted according to [36] with modifications and 3-deoxyanthocyanidins from sorghum using the [37] methodology. The material was peeled and cut into small pieces (2 cm^2^), and mixtures with alkaline water at pH 8, adjusted with sodium hydrogen carbonate (0.103 mg mL^−1^ NaHCO_3_) and sodium carbonate (0.008 mg mL^−1^ Na_2_CO_3_), were prepared with a solid–liquid ratio of 1:5 (*w*/*v*) for black carrots, and mixtures with acidified water at pH 3, adjusted with citric acid (0.961 mg mL-1, C_6_H_8_O_7_), were prepared with a solid–liquid ratio of 1:3 (*w*/*v*) for radish, sweet potato peels, and 1:1 (*w*/*v*) for sweet potato pulp.

Anthocyanins from black carrots, radish, and purple sweet potato were extracted in two different conditions: 1) using an alkaline aqueous solution at pH 8, adjusted with sodium hydrogen carbonate (0.103 mg mL^−1^ NaHCO_3_) and sodium carbonate (0.008 mg mL^−1^ Na_2_CO_3_) (extracts at pH 8), and 2) using an acidified water at pH 3, adjusted with citric acid (0.961 mg mL^−1^, C_6_H_8_O_7_) (extracts at pH 3). In both cases, the extraction mixtures were heated on a hotplate (Heidolph MR 3001, Schwabach, Germany) for 10 min and homogenized using an ultra-turrax T25 (Janke & Kunkel, Staufen, Germany) at 8000 rpm for 10 min. Subsequently, all the extraction mixtures were sonicated (ultrasonic processor, Hielscher UP200S, Berlin, Germany) at 50% amplitude and 0.5 cycles for 10 min.

Sorghum 3-DXA were extracted from sorghum leaves using two extraction methods: solvent extraction and ultrasound-assisted (UA) at room temperature. In both cases, the ground plant material was mixed with an alkaline aqueous solution at pH 10 in a 1:20 (*w*/*v*) solid to liquid ratio. For the solvent extraction, the mixture was placed on an orbital shaker (Unitronic-OR, Selecta, Barcelona) at 25 °C and stirred at 70 rpm for 30 min. In the ultrasound method, the extraction mixtures were sonicated (ultrasonic processor, Hielscher UP200S, Berlin, Germany) at 50% amplitude and 0.5 cycles for 10 min.

After the extraction step, all the extracts were filtered through a 150 mm diameter filter (Whatman No. 1 Qualitative) at reduced pressure (GAST DOA-P104-BN, Redditch, United Kingdom). The supernatants were frozen at −80 °C, freeze-dried (Scanvac Cool Safe, Labogene, Bjarkesvej, Denmark). The vacuum pressure of the freeze-drier was set at 0.2 hPa, the plate temperature was 20 °C, and the condenser was at −50 °C for 24 h. The powdered colouring matters were weighed and stored in glass bottles (Schott 250 mL, Stuttgart, Germany) inside desiccators containing silica gel, kept in the dark at room temperature. All the extractions were performed in duplicate.

### 2.4. Colouring Matter

#### Selection of the Better Extraction Method Based on UV-Vis Spectrophotometry Analysis

The concentration of anthocyanin (black carrots, radish, and purple sweet potato pulp and peel) or of 3-DXA (sorghum) in the powdered extracts was determined to select the better extraction method for each plant. The extracts were dissolved in aqueous solutions at pH 3 (for the extracts prepared at pH 3), pH 8 (for the extracts prepared at pH 8), and pH 10 (sorghum extracts). The extracts were prepared in different concentrations: 0.1% (*w*/*v*) for black carrots, sorghum, and commercial dye; 0.3% (*w*/*v*) for radish and purple sweet potato peel; and 0.6% (*w*/*v*) for purple sweet potato pulp. The absorbance of the solutions was measured using a double-beam UV-Vis spectrophotometer (double-beam; Hitachi U-2010, Santa Clara, CA, USA) between 400 nm and 700 nm. The aqueous solutions at pH 3, pH 8, or pH 10, utilized to dissolve the extracts, were used for baseline correction.

The concentration of anthocyanins in aqueous solutions was calculated using the Beer–Lambert law:(1)A=ε∗c∗l
where *A* is the absorbance, *ε* the molar absorptivity (extinction coefficient) of the anthocyanin compound at the specific wavelength (for Cyanidin-3-glucoside, typically 26,900 L.mol.cm^−1^ and for apigenidin, typically 27,629.5 L.mol.cm^−1^), *c* is the concentration of the anthocyanin solution in mol L^−1^, and *l* is the path length of the cuvette in cm. The plot of the absorbance versus wavelength was used to observe the absorption spectrum of the anthocyanin sample.

### 2.5. Characterization of the Plant Extracts

The extracts were characterized both chemically (structural characterization of anthocyanins and content of total anthocyanins and total phenolic compounds) and functionally (antioxidant and antidiabetic capacities). A commercial food colouring (dye factory) of unknown composition, used as a control in the formulation of colouring coatings, was also analysed.

#### 2.5.1. Chemical Characterization of the Plant Extracts

##### Analysis by HPLC-DAD-MS and UHPLC-HRMS/MS

The identification and structural characterization of anthocyanins present in the extracts was achieved by high-performance liquid chromatography (HPLC) coupled with mass spectrometry (MS) assays.

Aliquots of 10 µL of each extract were initially analysed on a LC-MS system, consisting of an HPLC Ultimate 3000RS with Diode Array Detector (DAD) coupled to an ion trap LCQ Fleet mass spectrometer (Thermo Scientific, Carlsbad, CA, USA), with an ESI source. The full scan acquisition was performed in the positive ion mode in the range 100–1500 *m*/*z*. The DAD system was monitored in the range between 220 and 700 nm, and additional UV/VIS spectra were recorded at 520, 320, and 280 nm.

The identification and structural characterization of anthocyanins were achieved with an UHPLC Elute system coupled to a QqTOF Impact II high-resolution mass spectrometer (Bruker Daltonics, Bremen, Germany), interfaced with an ESI source operating in the positive mode. Mass spectra were acquired in the data-dependent mode (DDA) in a mass range between 100 and 1500 *m*/*z*, with an acquisition rate of 3 Hz, using a dynamic method with a fixed cycle time of 3 s.

The anthocyanins were separated by a Kinetex C18 core–shell column (150 mm × 2.1 mm; particle size 2.6 μm; Phenomenex, Torrance, CA, USA), using an elution gradient of 0.1% *v*/*v* formic acid in water (mobile phase A) and methanol (mobile phase B), at a flow rate of 300 μL min^−1^, More experimental details can be found in [13].

##### Total Antocyanin Content

The total anthocyanin content (TAC) was determined by the pH differential method following the methodology described by [38]. This methodology involves measuring the absorbance of anthocyanin extracts at two different pH values (pH 1 and 4.5) and using these absorbance values to calculate the concentration of anthocyanins. Anthocyanin extracts were prepared at concentrations of 1.28 mg/mL for radish and sweet potato peel, 2.46 mg/mL for sweet potato pulp, and 0.63 mg/mL for black carrot, sorghum, and commercial colourant. The TAC was calculated using the equation:(2)TAC (mg/L)=(A×DF×MW)/(ε×l)
where *A* is the difference in absorbance between pH 1.0 and pH 4.5, *DF* is the dilution factor, *MW* is the molecular weight of cyanidin-3-glucoside (the anthocyanin standard 484.83 g mol^−1^) or the specific anthocyanin being analysed (in sorghum apigenidin, 255.24 g mol^−1^), *ε* is the molar absorptivity coefficient of cyanidin-3-glucoside and apigenidin depending on the plant in study, and *l* is the path length of the cuvette (1 cm).

##### Total Phenolic Compounds Content

The total phenolic content (TPC) of the extracts was estimated using the Folin–Ciocalteu colourimetric method, as described by [39] and modified by [40], with gallic acid as the standard phenolic compound. Diluted samples (1–3 mL) were added to 10 mL volumetric flasks containing distilled water and 0.5 mL of Folin–Ciocalteu reagent and shaken. After 5 min, 1.5 mL of a 200 g L^−1^ sodium carbonate solution was added, and the volume was adjusted to 10 mL with distilled water, mixed, and left to stand for 2 h. A blank reagent with distilled water was prepared. Absorbance was measured at 750 nm using a double-beam UV–visible spectrophotometer (Hitachi U-2010, USA). Concentrations of total phenolic compounds were determined based on the standard curve (y = 88.003x + 0.0288, r^2^ = 0.998) in terms of grams per liter of gallic acid equivalents (GAEs). The linearity range for this assay was 6.3 × 10^−4^−1.3 × 10^−2^ gL^−1^ GAE, with an absorbance range of 0.08–1.13 AU. All determinations were performed in triplicate.

#### 2.5.2. Biological Capacities of the Extracts

##### Antioxidant Capacity

Radical scavenging capacity assay (DPPH^•^)

The scavenging effect of the DPPH^•^ free radical was assessed spectrophotometrically by the modified method of [41]. A volume of 0.1 mL of each plant extract at various concentrations was added to 2 mL of 0.07 mmol L^−1^ DPPH^•^ in 950 g L^−1^ ethanol, followed by shaking and incubation for 60 min at room temperature in the dark. The absorbance was measured at 517 nm using a double-beam UV–visible spectrophotometer (Hitachi U-2010, USA). A blank sample with ethanol and DPPH^•^ solution served as the negative control. Antioxidant capacity results were expressed as trolox equivalents (TEs) based on a standard curve (y = −0.0011x + 0.0109, r^2^ = 0.999) with a linearity range of 23.97–800 µmol TE g^−1^ and an absorbance range of 0.037–0.85 AU. All determinations were performed in triplicate.

Ferric ion reducing antioxidant power

Ferric ion reducing antioxidant power (FRAP) measures the formation of a blue-coloured Fe^2+^-tripyridyltriazine compound from the rust-coloured oxidized Fe^3+^ form by the action of electron-donating antioxidants. The FRAP assay was performed using the modified methodology of [42]. FRAP reagent was prepared with 1 mmol L^−1^ TPTZ and 2 mmol L^−1^ ferric chloride in 0.25 mol L^−1^ sodium acetate at pH 3.6. Diluted extracts (200 µL in 500 g L^−1^ methanol) were mixed with 1.8 mL FRAP reagent, allowed to stand for 4 min at room temperature, and the absorbance of the blue complex was measured at 593 nm against a water blank using a double-beam UV–visible spectrophotometer (Hitachi U-2010, USA). A standard curve (y = 0.0213x + 0.0057, r^2^ = 0.998) was prepared using different concentrations of iron sulphate, with a linearity range of 1.25–50 µmol L^−1^ and an absorbance range of 0.02–1.03 AU. FRAP values are presented as µmol Fe^2+^ g^−1^ of sample. All determinations were performed in triplicate.

##### Antidiabetic Capacity

Antidiabetic capacity was assessed by evaluating the potential of the extracts to inhibit the carbohydrate digestive enzymes α-amylase and α-glucosidase, according to the procedures described by [43,44,45], respectively.

For the α-amylase inhibition assay, radish, black carrot, and purple sweet potato extracts were dissolved in water (20 μg mL^−1^) and then diluted to concentrations ranging from 2.5 to 18 μg mL^−1^. Sorghum extract was dissolved in 50% (*v*/*v*) ethanol:water (10 μg mL^−1^) and diluted to concentrations ranging from 2 to 9 μg mL^−1^. For the assay, 100 µL of α-amylase (0.5 mg mL^−1^ in sodium phosphate buffer 0.02 M, pH 6.9, 6.7 mM NaCl) and 100 µL of each extract concentration were incubated at 37 °C for 10 min. Then, 100 µL of starch suspension (1% *w*/*v* in sodium phosphate buffer 0.02 M, pH 6.9, 6.7 mM NaCl) were added, and the mixtures were further incubated at 37 °C for 10 min. After that, 200 μL of 3,5-dinitrosalicylic acid reagent (20 mL of 96 mM DNS, 8 mL of 5.315 M sodium potassium tartrate tetrahydrate in 2 M NaOH, and 12 mL of water) were added, and the tubes were boiled on a heating block (Bioblock Scientific Code 92607) at 100 °C for 15 min. Afterwards, the mixtures were cooled in an ice bath, 2 mL of distilled water was added, and the absorbance was read at 530 nm on a UV-Vis spectrophotometer (SPEKOL 1500, Analytik, Jena, Germany). The mixtures without plant extracts were used as negative controls and mixtures without α-amylase were used as samples’ blanks. Increasing concentrations of acarbose (5.0–100.0 µgmL^−1^) were used as positive controls.

The enzyme inhibition rate was calculated according to Equation (3):(3)$$\%\;Inhibition=\left(1-\left(\frac{{Abs}_{A}-{Abs}_{B}}{{Abs}_{CN}}\right)\right)*100$$
where *Abs_CN_* is the absorbance of the negative control, *Abs_A_* the absorbance of the samples, and *Abs_B_* the absorbance of the sample blanks. Values were assessed in triplicate, and the results were expressed as the final concentration (mg mL^−1^), in the reaction mixture, which reduces the enzyme activity by 50% (IC_50_). Data are presented as means ± standard deviations.

For the α-glucosidase assay, the extracts were first dissolved in water to a concentration of 5 mg mL^−1^ and then diluted to concentrations ranging from 0.2 to 2.0 mg mL^−1^. The α-glucosidase (6.25 units mL^−1^) and 4-nitrophenyl-α-D-glucopyranoside (2.5 mM) were prepared in 0.1 M sodium phosphate buffer at pH 6.9. Reaction mixtures containing 5 µL of α-glucosidase, 125 µL of phosphate buffer (pH 6.9, 0.1 M), and 20 µL of the different extract concentrations were prepared in a 96-well microplate (Greiner Bio-One, Rainbach im Mühlkreis, Austria) and incubated for 15 min at 37 °C. Then, 20 µL of 4-nitrophenyl-α-D-glucopyranoside was added, and the plates were incubated for an additional 15 min, at 37 °C. To stop the reactions, 80 µL of 0.2 M Na_2_CO_3_ was added in each well. The absorbance was measured in a microplate reader (FLUOstar^®^ Omega Plate Reader, BMG Labtech, Ortenberg, Germany) at 405 nm. Reaction mixtures without extracts were used as negative controls, and reaction mixtures without the enzyme were used as samples’ blanks. Increasing concentrations of acarbose (80–980 µg mL^−1^) were used as positive controls. The extracts’ inhibitory capacity was calculated according to Equation (5):(4)$$\%\;Inhibition=\frac{\left({Abs}_{CN}-\left({Abs}_{A}-{Abs}_{B}\right)\right)*100}{{Abs}_{CN}}$$
where *Abs_CN_* is the absorbance of the negative control, *Abs_A_* is the absorbance of the sample, and *Abs_B_* is the absorbance of the sample blanks. Values were assessed in quadruplicate, and the results were expressed as the final concentration (mg/mL) in the reaction mixture, which reduces the enzyme activity by 50% (IC_50_). Data are presented as means ± standard deviations.

### 2.6. Confectionery Colouring Coatings

The methodology for the formulation of colouring coatings, based on anthocyanins and 3DXAs, to be applied to confectionery products, was adapted from the factory manufacturing process and involved the selection of purple colouring extracts (radish, black carrot, purple sweet potato, and sorghum) and polymeric materials (locust gum and Arabic gum) for application in sugar syrup. The concentrations of the colouring materials were adjusted according to the compatibility and desired characteristics of the chocolate almonds (Table 1). The control was a commercial colourant (Dye factory) of unknown composition. The quantity of colouring materials was selected, based on the same absorption (Abs = 1), at 510 nm for anthocyanins and 420 nm for 3-DXAs, in scanning the UV-Vis spectrum. The colouring materials were added to the polymeric materials and to the sugar syrup at 50 °C. The solutions were homogenized in the ultra-turrax (T25, Janke & Kunkel) at 8000 rpm for 5 min.

### 2.7. Thermostability of Confectionery Colouring Coatings

The stability of the formulated colouring materials was evaluated at a temperature of 90 ± 5 °C for 75 min. Samples (7 mL) of the formulations indicated in Table 1 were placed into test tubes (20 mL) and sealed hermetically. A commercial dye was used as a control in the temperature stability tests. The tubes were placed in the oven (Ventilcell 111, MMM Group), and duplicates were removed every 15 min and immediately cooled in an ice bath to stop thermal degradation. The changes in the total anthocyanins and 3-DXA contents during heat treatment were determined by measuring the absorbance at 510 and 520 nm, respectively, using a UV–visible spectrophotometer (dual beam; Hitachi U-2010). Assays were performed in duplicates.

The absorbance measurements of the colouring coating formulations were used to obtain the kinetic parameters according to [13] to estimate the thermostability of the anthocyanins and the 3-DXAs. The first-order reaction rate constants (K_d_), half-lives (t_1/2_), i.e., the time that is necessary for degradation of 50% of anthocyanins and 3-DXAs, were calculated by the following Equations (5) and (6):(5)$$\ln\left(\frac{C_{t}}{C_{0}}\right)=-k\times t$$
(6)$$t_{1/2}=-ln(0.5)\times k^{-1}$$
where *C*_0_ is the initial concentration of the anthocyanins or 3-DXAs, and *C_t_* was the anthocyanins or 3-DXAs concentration after t minutes at 90 °C.

The integrated rate equation for a second-order reaction is determined from experimental data. It can be calculated from the initial rate of reaction and the initial concentrations of the anthocyanins or 3-DXAs. The rate equation is represented as:(7)$$\frac{1}{\left[A\right]t}=kt+\frac{1}{\left[A\right]0}$$
where *k* is the rate constant and [*A*]*_t_* is the concentration of the anthocyanins or 3-DXAs after t minutes of heating at 90 °C, and [*A*]_0_ is the initial concentration of the anthocyanins or 3-DXAs.

### 2.8. Purple Confectionery Coatings: Effects of Blue Dye Addition

The methodology for the formulation of purple-coloured polymeric coatings, was adapted from the factory manufacturing process, and incorporating anthocyanins and 3-DXAs extracts, were developed with and without the addition of spirulina as a blue dye (Table 1 and Table 2, respectively). These coatings were meticulously applied onto white solid sugar pastes, each cut into 5 cm^2^ squares, following the procedure outlined in Section 2.6. The application involved brushing the pastes three times at 30 min intervals, followed by drying in an oven set at 30 ± 5 °C. Subsequently, the samples underwent a 72 h drying period under the same temperature conditions.

### 2.9. Colourimetry

The colour of the coating pastes was assessed using colourimetry and compared against purple Easter almonds dyed with clean label colourants and those with synthetic dyes. All experiments were conducted in duplicate.

The colourimetric analysis was performed according to [13]. The colour properties in the CIELAB system, including lightness (L*), red-green chromaticity (a*), and yellow-blue chromaticity (b*), were measured in liquid samples using a Chroma Meter CR-400 (Konica Minolta, Japan), with water as the standard reference. For solid samples, colour measurements were determined with a CS-5 CHROMA SENSOR spectrophotometer (Datacolour International) using a 45/0 geometry, D65 illuminant, 10° angle, and including the specular component, with the CIELAB colour white as the standard reference. The colour difference between the samples was calculated using the equation:(8)$$\Delta Eab=\sqrt[{2}]{({\Delta L*)}^{2}+{\left(\Delta a*\right)}^{2}+{\left(\Delta b*\right)}^{2}}$$
where ΔL*, Δa*, Δb* represent the difference between each parameter for the anthocyanin and 3-DXA colourants. The measurements were performed in triplicate.

### 2.10. Statistical Analysis

The results were submitted to one-way analysis of variance (ANOVA) using multiple comparison tests (Tukey HSD) to identify differences between groups. Statistical analyses were tested at a 0.05 level of probability. The range, mean, and relative standard deviation (RSD) of each parameter were calculated using the software, StatisticaTM 12.0 [46].

## 3. Results and Discussions

### 3.1. Selection of the Extraction Method

Figure 1 shows the total anthocyanin content (TAC) obtained for plant matrix extracts at pH 3 and pH 8 after scanning the absorption spectrum at 530, 546, and 580 nm wavelengths, corresponding to red, pink, and purple colours, respectively.

For both pH values, PP extracts exhibited the highest TAC, while PSP showed the lowest. Significant differences (*p* > 0.05) were observed in TAC for all samples at pH 3 (Figure 1), as well as at pH 8, excluding for PPS. Extraction at acid pH resulted in higher TAC for all samples extracts, with the exception of the BC extract.

The TAC values reported by [15] for aqueous extracts of dried sweet potato pulp are lower than those presented in Figure 1. This discrepancy may be due to the use of ultrasound-assisted (UA) methods, which can cause the cell walls of sweet potato pulp and peel to burst, thereby releasing more anthocyanins.

Ref. [47] found that extracting anthocyanins from R peels with acidified hexane is more efficient than using citric-acid-acidified aqueous extraction. However, hexane is harmful to human health [48].

Although [19] observed that anthocyanin extraction at acidic pH is more stable than at alkaline pH, they obtained lower TAC values and redder colour. In contrast, extraction at alkaline pH resulted in a colour closer to purple.

Figure 2 presents the values of 3-DXA obtained from sorghum leaves using two extraction methods: solvent extraction at room temperature and ultrasound-assisted (UA). The results indicate a higher 3-DXA content with the UA extraction method. [24] reported similar findings, suggesting that the use of UA causes the sorghum cell walls to burst, releasing more 3-DXA. However, an aqueous solution at pH 9 did not produce a purple colour, as noted by [12]. These discrepancies may be attributed to differences in the solvents used, extraction methods, and vegetable varieties, as well as varying soil and climatic conditions. These factors influence the plants and lead to different secondary metabolite profiles.

Based on the results obtained, the following methods were selected for preparing the final colouring extracts: ultrasound-assisted (UA) extractions at pH 3 for all samples, with the exclusion of BC and S with the extractions performed at pH 8 and pH 10, respectively. These final extracts were chemically and functionally characterized and applied in the development of colouring coatings in confectionery.

### 3.2. Chemical Characterization of the Final Extracts

#### 3.2.1. Anthocyanin and Phenolic Compound Content

The total anthocyanin content (TAC) of the plant extracts (Table 3) showed significant differences (*p* < 0.05). The R extract had the highest TAC (4.54 ± 0.10 mg g^−1^), while S had the lowest (0.39 ± 0.05 mg g^−1^). These results are lower than those reported by [15] for sweet potato extracts (5.23 ± 0.07 mg cyanidin-3-glucoside g^−1^). The authors of [49] found 3-DXA values between 13.7 and 35.5 mg cyanidin-3-glucoside g^−1^ in methanolic extracts. However, these values are not directly comparable to those in Table 3, as 3-DXA was calculated using the molar extinction coefficient (ε) of apigeninidin, the main identified compound.

For BC, the TAC values are lower than those reported in previous studies, such as [50], who obtained 0.18 mg cyanidin-3-glucoside g^−1^ using aqueous extraction. The observed differences are due to the different extraction solvents and the use of the molar extinction coefficient (ε) of cyanidin-3-glucoside to calculate TAC.

The total phenolic content (TPC) of the plant extracts is shown in Table 3. The dye factory (DF) had the highest TPC (74.5 ± 3.7 mg GAE g^−1^), while PSP had the lowest (6.78 ± 0.16 mg GAE g^−1^). No significant differences (*p* < 0.05) in TPC were observed for the PSP and PP extracts or for the BC and S extracts. These results are consistent with those reported by [15], who found TPC values of 12.81 ± 0.01 mg GAE g^−1^ in acidified aqueous sweet potato extracts.

The TPC results for S are similar to those reported by [51], which ranged from 7.30 to 107.85 mg GAE g^−1^ using acid extraction. The TPC values obtained for R peel using enzymatic extraction corroborate those in the literature, with [52] reporting TPC values between 40.27 and 74.78 mg GAE g^−1^. The authors in [53] used methanol and acetone for extraction from radish peel, obtaining lower TPC values (1.7 and 39.4 mg GAE g^−1^).

#### 3.2.2. Identification and Structural Characterization of Anthocyanins in the Colouring Extracts

The structures of anthocyanins were determined using a combination of HPLC-DAD and high-resolution tandem mass spectrometry (HRMS/MS) analyses.

For the identification of anthocyanins, UV-VIS spectra obtained from HPLC analysis with a diode array detector (DAD) provide significant information [54]. A typical UV-VIS spectrum of an anthocyanin presents two characteristic absorption bands, one at 260–280 nm (UV region) and the other at 490–550 nm (VIS region). If sugar residues are acylated with hydroxycinnamic acids, an additional band in the range of 310–340 nm is found. This band provides information regarding the acylating aromatic acid. An λ_acyl_ in the range of 320–339 nm indicates sinapic or ferulic acid, whereas an λ_acyl_ around 310–315 nm indicates *p*-coumaric acid. Additionally, a hump at 400–450 nm is also observed, whose intensity depends on the number of sugar moieties attached to the anthocyanidin moiety.

The detection of anthocyanins by ESI-MS was achieved in a positive ionization mode, since anthocyanins are primarily present as flavylium cations [M]^+^ under acidic mobile phase conditions. Due to the lack of reference standards, compounds were tentatively identified based on accurate mass measurements with a high level of accuracy (mass error < 5 ppm; mSigma < 25), combined with molecular formulae as well as by comparison of UV-VIS data and fragmentation patterns with the literature data.

Figure 3 illustrates the HPLC-DAD chromatograms of colouring extracts obtained between 450 and 530 nm. The identified anthocyanins, their retention times, UV–VIS absorption maxima, and mass spectra characteristics are presented in Appendix A. The results indicate that, except for sorghum, most anthocyanins are present in the acylated form.

The chromatogram profiles of the PSP and PP extracts, Figure 3a,b, clearly indicate that they both possess an identical composition of anthocyanins. MS screening enabled the identification of thirteen anthocyanins, two of which correspond to a non-acetylated form. Based on the MS/MS results, it was concluded that all anthocyanins belong to the cyanidin (*m*/*z* 287) and peonidin (*m*/*z* 301) families, with the anthocyanidin 3-sophoroside-5-glucoside as a common structure. As previously reported, these compounds display variable substitution with hydroxycinnamic acids attached to the C3 of the flavylium ring, in accordance with previously reported results [55,56,57,58,59,60].

The major anthocyanins in the black carrot extract (Figure 3c) are also cyanidin-based, containing different sugar moieties, non-acylated or acylated with *p*-coumaric, ferulic, and sinapic acids. Therefore, a local maximum at 330 nm was observed in their DAD chromatograms. A series of minor compounds that elute at higher retention times were assigned to acylated peonidin and pelargonidin anthocyanins. The tandem mass spectra displayed characteristic fragment ions attributed to the presence of a trisaccharide (one pentose and two hexoses) which is acylated with the hydroxycinnamic acids. Cyanidin-3-xylosylglucosyl-galactosyl acylated with ferulic acid (t*_R_* 8.20 min) is the predominant anthocyanin in purple carrots [61,62,63,64].

The HPLC-DAD chromatogram recorded at 520 nm for the extract of radish peel (R) e is presented in Figure 3d. The absorption spectra of anthocyanins were observed at shorter wavelengths compared to those observed for the anthocyanins from the other extracts. This blue shift indicates that the R extract has pelargonidin derivatives. MS analysis enabled the identification of sixteen pelargonidin-based compounds that were di-glycosylated at the C-3 and glycosylated at the C-5 positions. These compounds primarily comprised one or two hydroxycinnamoyl groups on the glycosylated moiety at the C-3 position (feruloyl, caffeoyl, and *p*-coumaroyl) and malonyl groups on the hexose at the C-5 position, as previously described for extracts of red radish (*Raphanus sativus* L.) [65,66,67,68].

The UV-VIS spectrum of sorghum (Figure 3e) revealed a peak with an absorbance in the blue zone (λ_max_ 472 mn) that in the mass spectrum yielded a cation with *m*/*z* 255.0660, which was assigned to apigeninidin. This is a primary pigment in sorghum, which belongs to the group of 3-desoxy-anthocyanidins [13].

The profile of anthocyanins from dye factory extract (DF) is shown in Figure 3f. Based on MS results, it was concluded that the extract only contains acylated derivatives of cyanidin and peonidin. A series of peaks eluting between 8.44 and 10.23 min revealed a similar fragmentation pattern, with the loss of a glycosyl residue (162 u) followed by the loss of a cinnamyol-sophoroside residue, leading to the aglycone moiety. These findings suggest the presence of anthocyanin-3-(cinnamoyl)sophoroside-5-*O*-glucoside derivatives, which correspond to the anthocyanins described in PSP and PP extracts. At higher retention times, four other acylated cyanidins were identified. The tandem mass spectra show that these precursor cations fragment by the loss of a malonoyl-glycosyl residue (248 u) followed by a diacylated sophoroside residue, resulting in the cyanidin fragment ion. It seems that the presence of a malonic acid attached to the C5-position is characteristic of red radishes, a widespread vegetable occurring in different shapes and varieties. According to published data, the type of anthocyanin in radish depends largely on the variety studied. Some varieties only contain acylated anthocyanins, such as cinnamoyl, malonyl, and its derivatives. The primary aglycone in the red genotypes is pelargonidin, while cyanidin is mostly found the purple radish root [68,69]. The results indicate that the DF extract should consist of a mixture of purple sweet potato and purple radish root.

#### 3.2.3. Biological Capacities: Antioxidant and Antidiabetic

The biological capacities of the extracts were evaluated for antioxidant capacity using the DPPH^•^ and FRAP methods, and the antidiabetic capacity was evaluated through the inhibition of the enzymes ɑ-amylase and ɑ-glucosidase; the results are represented in Table 4.

The antioxidant capacity of the extracts was assessed by the DPPH^•^ assay, which evaluates the radical scavenging capacity, and by the FRAP assay, which evaluates the reducing capacity. All the plant extracts presented antioxidant capacity detected by both assays.

The antioxidant capacities of the plant extracts differed significantly (*p* < 0.05) between the DPPH^•^ and FRAP methods. In the DPPH^•^ method, no significant differences (*p* > 0.05) were observed between the extracts from radish, sweet potato peel, and sorghum. However, the FRAP method revealed significant differences (*p* < 0.05) between these extracts.

The PP extract exhibited the highest antioxidant capacity in the DPPH^•^ method (0.37 ± 0.008 mmol TE g⁻^1^ DW), while the DF extract showed the lowest (0.056 ± 0.01 mmol TE g⁻^1^ DW). The results can be explained by the varying abilities of anthocyanins with different acylation patterns (S1) to donate hydrogen atoms or electrons, which affect their performance in the DPPH^•^ assay. In the FRAP assay, the DF extract demonstrated the highest antioxidant capacity (872.80 ± 13.35 mmol Fe^2^⁺ g⁻^1^), whereas the S extract had the lowest (67.89 ± 2.72 mmol Fe^2^⁺ g⁻^1^). This trend was consistent with the phenolic compound content in the DF and S extracts (Table 3), as phenolic compounds can donate electrons to reduce ferric ions to ferrous ions, resulting in higher antioxidant capacity in the FRAP assay.

The DPPH^•^ results for sweet potato peel are consistent with those reported by [70], ranging from 150 ± 0.02 mg TE g^−1^ DW in the freeze-dried extract. Notably, DPPH^•^ values for sweet potato peel extracts were not found in the existing literature.

For sorghum extracts, the FRAP method results (Table 4) with alkaline solutions were lower than those described by [51], who used acidic solvents.

The antioxidant capacity results obtained using both DPPH^•^ and FRAP methods for radish and purple carrot were superior to those reported in the literature [52,71,72]. These discrepancies may be attributed to the different solvents and concentrations used in the tests and the type and structure of anthocyanin compounds, identified in Section 3.2.2, including their acylation patterns, that play a crucial role in determining the antioxidant capacity as measured by DPPH^•^ and FRAP methods. The variations in these compounds across different plant matrices can lead to differences in observed antioxidant capacities.

The control of the postprandial hyperglycemia is a major factor in the management of type 2 diabetes [73]. Thus, preventing the absorption of glucose by the inhibition of the carbohydrate-hydrolysing enzymes α-amylase, responsible for dietary starch digestion, and α-glucosidase, responsible for the disaccharide’s hydrolysis to glucose, could potentially prevent type 2 diabetes and its complications [73,74,75]. Indeed, acarbose and other inhibitors of the carbohydrate digestive enzymes are commonly used as oral drugs to control type 2 diabetes [76].

All the extracts inhibited α-amylase, but, under the conditions assayed, only sorghum extract was able to inhibit α-glucosidase capacity (Table 4). Significant differences (*p* < 0.05) were observed between the α-amylase inhibitory capacity of the plant extracts, with IC_50_ values ranging from 1.40 mg mL^−1^, for sorghum extract, to 3.70 mg mL^−1^, for black carrot extract. Although all extracts showed α-amylase inhibitory capacity, they were all less effective than acarbose (positive control).

Results obtained with sorghum extract agree with previous studies which reported both α-amylase and α-glucosidase inhibitory capacities of 70% ethanolic extracts from different sorghum grain genotypes [73,76]. In those studies, and in agreement with the present results, sorghum extracts also presented an α-glucosidase inhibitory capacity stronger than acarbose. Additionally, [77] also reported the α-glucosidase inhibitory capacities of methanolic extracts of bran and flour of six sorghum varieties.

Other authors also described a more pronounced α-amylase than α-glucosidase inhibitory capacity for aqueous radish extract [75], freeze-dried purple carrot extracts [78], and 12% ethanolic extracts of purple sweet potato [79].

It is known that several polyphenols can inhibit α-amylase and α-glucosidase enzymes, due to their ability to bind amino acid residues at enzyme active sites [76,77]. However, in the present work, the capacity of the plant extracts to inhibit the carbohydrate-hydrolysing enzymes correlated poorly with both total phenolic or total anthocyanin contents (Table 3 and Table 4). Thus, in agreement with previous studies, our results also suggest that, for the inhibition of digestive enzymes, the type of polyphenols is more significant than the total phenolic amounts [80]. Thus, polyphenols with high molecular weight and degree of polymerization seem to present a stronger α-glucosidase inhibitory capacity. Other factors, like the class and subclasses of the phenolic compounds or the type of sugar units linking the phenolic structures, could also play pivotal roles in the inhibitory capacities of polyphenols. Moreover, the diversity of phenolic compounds present in plant extracts may synergistically and/or antagonistically contribute to the α-amylase and α-glucosidase inhibition [73,79].

The α-amylase and, particularly, the α-glucosidase inhibitory capacity of sorghum extract suggest that its incorporation into foods may be a promising strategy to delay carbohydrate digestion and prevent postprandial hyperglycemia.

### 3.3. Development of Colouring Coatings in Confectionary

#### 3.3.1. Thermostability of Colouring Coatings in Confectionery

The results for the degradation kinetics of coloured coatings in syrup sugar formulations with anthocyanins and 3-DXA are presented in Table 5. The data indicate that longer exposure times to temperature result in shorter half-lives (t_1/2_) for anthocyanins and 3-DXA, along with higher degradation constants (k_d_).

The temperature resistance of the plant extracts differed significantly (*p* < 0.05) in terms of degradation constants (k_d_) and half-lives (t_1/2_). No significant differences (*p* > 0.05) were observed in degradation constants (k_d_) between the radish extracts with both gums (R-L and R-AG), the dye factory coating with locust bean gum (DF-L), and sweet potato extracts without coating (PP-UC and PSP-UC). However, for locust bean gum, the half-lives (t_1/2_) revealed significant differences (*p* < 0.05) between the dye factory (DF) and sorghum (S) extracts.

Regarding the use of gum Arabic and locust bean gum to stabilize anthocyanins and 3-DXA extracts, locust bean gum demonstrated higher temperature resistance over time for the extracts: S (k_d_ = 72.83 ± 7.04), DF (k_d_ = 21.55 ± 1.08), BC (k_d_ = 15.7 ± 1.38), and R (k_d_ = 1.6 ± 0.27). In contrast, gum Arabic provided better stabilization for sweet potato peel (PP) (k_d_ = 12.87 ± 1.21) and pulp (PSP) (k_d_ = 4.61 ± 2.60). For both gum coatings, radish extracts exhibited the lowest temperature resistance. However, dye factory (DF) extracts without gum coatings showed the highest temperature stability in the syrup sugar formulations (k_d_ = 98.83 ± 0).

This increased stability may be due to the self-association of anthocyanins, which strengthens the interaction between them. The DF extracts contained a mixture of anthocyanins originating from purple sweet potato and purple radish root. Furthermore, the stability is also enhanced by hydrogen bonds between the sugars linked to the anthocyanins. The position and size of the sugars are crucial for the overall alignment and stability of the complex [9].

Ref. [81] reported higher stability (t_1/2_ = 96 h) for sweet potato peel extracts at pH 3 and 80 °C, compared to the values in Table 6. This increased stability can be attributed to the reduced stability of anthocyanins in sucrose, which forms degradation products such as fructose, thereby decreasing anthocyanin stability [82]. Conversely, [83] corroborated the data obtained for the thermostability of purple sweet potato pulp at 80 °C, reporting a half-life of 8.5 ± 2.2 h for the same dye.

Regarding black carrot anthocyanins, [84] demonstrated high stability at pH 1 and significant stability at pH 8 and 90 °C for 75 min. The increased stability at pH 8 can be attributed to the protective effect of polysaccharide coatings on anthocyanin stability, primarily through hydrophobic interactions within the anthocyanin–polysaccharide complex. These interactions protect the flavylium cation from nucleophilic attack by water, thereby preventing the formation of the colourless hemiketal form [85]. The type of acylated anthocyanins also influences temperature stability due to intramolecular stacking interactions [86].

In all extracts, the predominant anthocyanin was acylated, though with different molecular groups. In the BC extract, cyanidin-based anthocyanins were di-acylated with p-coumaric, ferulic, and sinapic acids, with minor amounts of peonidin and pelargonidin anthocyanins acylated with cinnamic acids. PSP and PP extracts revealed the presence of cyanidin di-acylated anthocyanins containing hydroxycinnamic acids. The R extract contained pelargonidin derivatives di-glycosylated with hydroxycinnamoyl groups and glycosylated with feruloyl, caffeoyl, and p-coumaroyl. The DF extract appeared to be a mixture of PP, PSP, and R.

Overall, the increased stability of di-acylated anthocyanins compared to mono-acylated anthocyanins can be attributed to their enhanced intramolecular stacking and protection against environmental stressors such as heat.

Coloured coatings with sorghum extracts showed better temperature resistance in both locust and Arabic gum compared to the dye factory, particularly at alkaline pH. This aligns with [13], who also reported that sorghum stability is better at alkaline pH than acidic pH.

Figure 4 illustrates the degradation percentages of coloured coatings with extracts from purple sweet potato (PSP—pulp; PP—peel), black carrot (BC), radish (R), sorghum (S), and dye factory (DF) in Arabic gum (AG) or locust gum (L), as well as uncoated (UC) extracts in sugar syrup following temperature stability tests. Different letters in the same column indicate significant differences (*p* < 0.05).

Significant differences (*p* < 0.05) were observed for R and PP and PSP extracts coated with Arabic gum and locust gum compared to other coloured coatings using the same polymeric materials.

The lowest dye degradation percentage was achieved with locust-gum-based coatings for S extracts and DF, and the highest degradation was found in R extracts. Arabic gum coatings exhibited the same trend as locust-gum-based coatings. Among uncoated extracts, the DF showed the lowest degradation, while R had the highest.

Locust gum provided better temperature resistance for anthocyanins than Arabic gum. This is attributed to the hydrophilic nature of anthocyanin molecules, which form weak interactions with the hydrophobic chains of the Arabic gum glycoprotein, as reported by [87]. Additionally, an increase in Arabic gum concentration from 0.5% to 1.5% may enhance the stabilization of anthocyanins during thermal processing, as demonstrated by [88] with polymeric coatings of black carrot dyes using 1.5% gum Arabic.

Figure 5 presents the total colour variation in the coloured coatings in thermostability tests. The results indicate that S extracts coated with both gums exhibited the highest colour variation. The BC extract showed the lowest colour variation when coated with Arabic gum, while DF extracts had better stabilization with locust bean gum coatings. Among the colourant extracts without polymeric coatings, the S extract also displayed the highest colour variation, whereas the factory dye extract showed the smallest variation.

There was no significant difference (*p* < 0.05) between the BC colour extracts coated with Arabic gum and the extracts of R, BC, PP, and DF coated with locust bean gum, as well as the extracts of R, BC, PP, and DF without polymeric coatings. These results suggest that locust bean gum provides similar colour stabilization to the extracts without any coating.

#### 3.3.2. Purple Confectionery Coatings: Effects of Blue Dye Addition

Purple-coloured coatings in sugar pastes

Figure 6 shows the colour variation (ΔEab*) observed in the extracts S, PSP, PP, R, BC, and the dye DF (used as a control in sugar pastes) when applied with polymeric coatings on white sugar pastes. PSP extracts exhibited less variation compared to almonds coloured with DF (designated dye factory clean label: DF-CL) and SD synthetic dye. In contrast, the DF extract exhibited the greatest colour difference. Notably, sorghum extracts showed a significant difference in colour variation between almonds coloured with the two types of dyes.

Furthermore, statistically significant differences (*p* < 0.05) were evident for the coloured extracts and those applied to sugar pastes in all dyes when compared with coloured almonds from the factory.

Purple-coloured coatings in sugar pastes, with the adding of spirulina blue dye

Figure 7 shows the colour variation (ΔEab*) observed in the extracts S, PSP, PP, R, BC, and in the dye DF (used as a control) when mixed with spirulina dye and applied with polymeric coatings on sugar pastes. Notably, sweet potato pulp extract exhibited the most subtle colour variation in polymeric-coated sugar pastes with added blue dye, both in almonds coloured with the factory dye DF-CL and the synthetic dye (SD). Conversely, the DF dye extract produced the most pronounced colour discrepancy.

Statistically significant disparities (*p* > 0.05) in colour were observed between extracts enriched with spirulina blue dye on all dye substrates, except for PSP extracts applied to sugar pastes, when compared with almonds coloured with the DF-CL dye.

A comparative analysis of the colour variation between the colouring extracts, with and without the inclusion of the blue dye (Figure 6 and Figure 7), versus almonds coloured with the DF-CL dye, revealed less colour variation with the incorporation of spirulina blue dye. Additionally, the five-month storage test demonstrated a reduction in colour variation compared to DF-coloured almonds, with the colour remaining stable throughout the storage period.

This result allows us to conclude that the PSP extract is more suitable than the DF dye for colouring almonds purple and replacing synthetic blue dyes, because it is closer to the purple colour with the addition of blue spirulina dye.

### 3.4. Principal Component Analysis (PCA)

The eigenvalues and the extraction of the main components are presented in Table 6, where the first two components explain 60.34% of the variance: the first component explains 38.07%, and the second component explains 22.27%. Principal components with eigenvalues greater than 1 are considered significant (*p* < 0.05). In this analysis, the eigenvalues for the first two principal components are 3.43 and 2.00, respectively.

**Table 6 foods-13-02450-t006:** Eigenvalues and extraction of the principal components.

Principal Component 1	Eigenvalue	% Total	Cumulative	Cumulative
		Variance	Eigenvalue	%
1	3.43	38.07	3.43	38.07
2	2.00	22.27	5.43	60.34
3	1.30	14.48	6.73	74.82
4	0.91	10.11	7.64	84.93
5	0.73	8.07	8.37	93.00
6	0.35	3.89	8.72	96.89
7	0.17	1.87	8.89	98.76
8	0.10	1.16	8.99	99.92
9	0.01	0.08	9.00	100.00

The relationship between the variables (original parameters) and the principal components is determined by the correlation coefficients shown in Table 7.

Figure 8 and Figure 9 show the projection of the attributes onto the plane defined by the first two main components. The first principal component (PC1) presents a positive correlation with the degradation constants of locust gum (L-K_d_, 0.89) and Arabic gum (AG-K_d_, 0.87) and a negative correlation with the total phenolic content (TPC, −0.83), the degradation constant of uncoated dyes (SR-K_d_, −0.75), and the total anthocyanin content (−0.65). This indicates that as the gum degradation constants increase, the TPC, SR-K_d_, and anthocyanin content decreases.

The second principal component (PC2) is highlighted by a positive correlation with the total colour variation in the uncoated dyes (SR-Delta E, 0.84) and the half-life time of the Arabic gum coated dyes (AG-T_1/2_, 0.65). On the other hand, it presents a negative correlation with the total colour variation in coatings with Arabic gum (AG-Delta E, −0.82), the half-life time of dyes with locust gum (L-T_1/2_, −0.65), and antioxidant capacity (DPPH^•^, −0.69). This suggests that as the total colour variance of uncoated dyes and the half-life time of Arabic gum coated dyes increase, the total colour variance of Arabic gum coatings, the half-life time of locust gum dyes, and antioxidant capacity decrease.

The projection of the attributes onto the plane defined by the two main components and their respective samples is illustrated in Figure 8 and Figure 9. This projection results in three distinct groups according to the type of dye. The group containing the dyes PP, BC, and PSP presents similar degradation constants for both locust gum (L-K_d_) and Arabic gum (AG-K_d_) indicating consistent performance in maintaining dye stability through these coatings.

The group containing dye S presents greater total colour variation for both locust gum (L-Delta E) and gum Arabic (AG-Delta E) coatings. Dye S also has higher half-life values for locust gum (L-T_1/2_) and gum Arabic (AG-T_1/2_) coatings. This suggests that dye S is more susceptible to colour changes but maintains its stability for a longer period compared to other dyes.

The PP, BC, and PSP dyes show greater similarity with the factory dye (control) when considering the fourteen parameters analysed. This indicates that these dyes are good choices for use in coatings such as locust gum and Arabic due to their stability and colour retention properties.

Dye S, although stable for a longer period, is more prone to significant colour variations. These conclusions may be useful for specific coating applications based on dye performance characteristics.

## 4. Conclusions

The extraction at acidic pH yields higher total anthocyanin content for most plant extracts, except for black carrot and sorghum, while extraction at alkaline pH produces a more purple colour but less stable anthocyanins.

The total anthocyanin content and total phenolic content of plant extracts varied significantly, with ultrasound-assisted extraction at acidic pH yielding the highest values for radish extracts.

The plant extracts exhibited significant antioxidant capacities and enzyme inhibitory capacities, with variations influenced by the type of assay and polyphenol composition, indicating that sorghum extract, in particular, holds potential for managing postprandial hyperglycemia and type 2 diabetes.

The UV-VIS spectra obtained from HPLC-DAD conjugated with tandem mass spectrometry data, in positive ESI mode, demonstrated that the majority of anthocyanins in the extracts are acetylated, with the exception of sorghum, which contains apigeninidin, a primary pigment belonging to the 3-DXA group.

HPLC clearly demonstrated quantitative, but not qualitative, differences between the extracts of the peel and pulp of sweet potatoes. Thirteen anthocyanins, consisting of non-acylated or acylated peonidin and cyanidin glycosides, were identified. The major anthocyanin is a peonidin 3-caffeoyl-p-hydroxybenzoyl sophoroside-5-glucoside.

Extracts of black carrot contained five cyanidin-3-xyloslyl-glucosyl-galactosides, three of which are mono-acylated with ferulic, sinapic, and p-coumaric acids. The variety analysed also had traces of acylated peonidin and pelargonidin anthocyanins.

The extract of radish peel predominantly contained pelargonidin derivatives, as indicated by a blue shift in UV-VIS spectra. Sixteen pelargonidin-based compounds were identified, with the presence of one or two cinnamoyl groups on the di-glycosylated fraction at the C3 position and a malonyl at hexose at the C5 position.

The extract of dye factory showed cyanidin and peonidin acylated derivatives, with fragmentation patterns suggesting the presence of anthocyanin-3-(cinnamoyl) sophoroside-5-glucoside derivatives, similar to those in the peel and pulp of purple sweet potato extracts.

Locust bean gum provided better temperature resistance and lower degradation percentages for coloured coatings of plant extracts compared to Arabic gum, making it more effective for stabilizing anthocyanins during thermal processing.

Thermostability tests revealed that sorghum extracts exhibited the highest colour variation when coated with gums, while black carrot extracts coated with Arabic gum and dye factory extracts coated with locust bean gum showed the best colour stability, with locust bean gum providing comparable stabilization to extracts without any coating.

In sugar pastes, sweet potato pulp extracts exhibited less colour variation than dye factory and synthetic dyes when used alone and, when combined with spirulina blue dye, provided a more stable and subtle colour variation in polymeric-coated sugar pastes, making it a superior and more consistent choice for colouring almonds purple.

Future work could investigate whether anthocyanin-based coatings retain their biocapacity when applied to confectionery products. Conclusions may include findings on the ability of coatings to preserve antioxidant levels and potential health benefits in final products.

## Figures and Tables

**Figure 1 foods-13-02450-f001:**
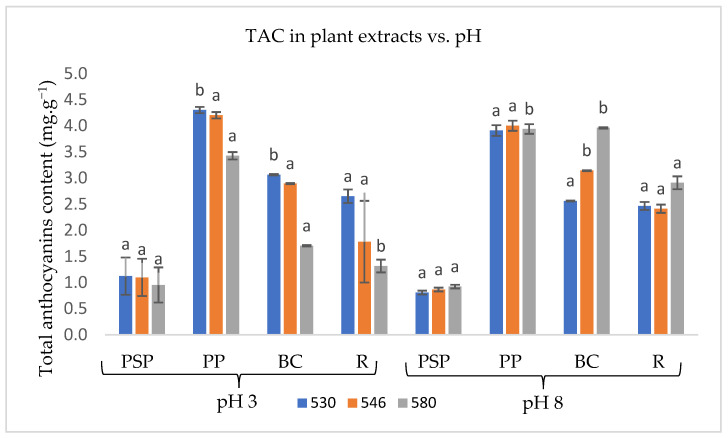
TAC for extracts of sweet potato pulp (PSP), sweet potato peel (PP), radish peel (R), and black carrot (BC) at pH 3 and pH 8, obtained at 530 nm, 546 nm, and 580 nm. Values are the average ± standard deviation of two parallel experiments. For each wavelength, different letters above the bars indicate significant differences (*p* < 0.05).

**Figure 2 foods-13-02450-f002:**
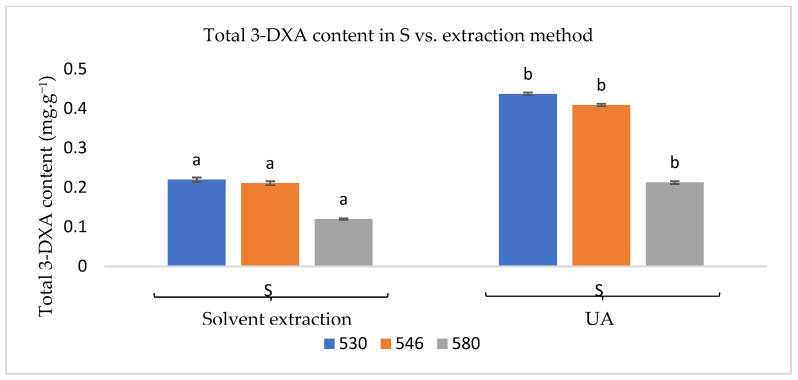
Total 3-DXA content in sorghum extracts at pH 10, measured at 530, 546, and 580 nm. Values are the average ± standard deviation of two parallel experiments. For each wavelength, different letters above the bars indicate significant differences (*p* < 0.05).

**Figure 3 foods-13-02450-f003:**
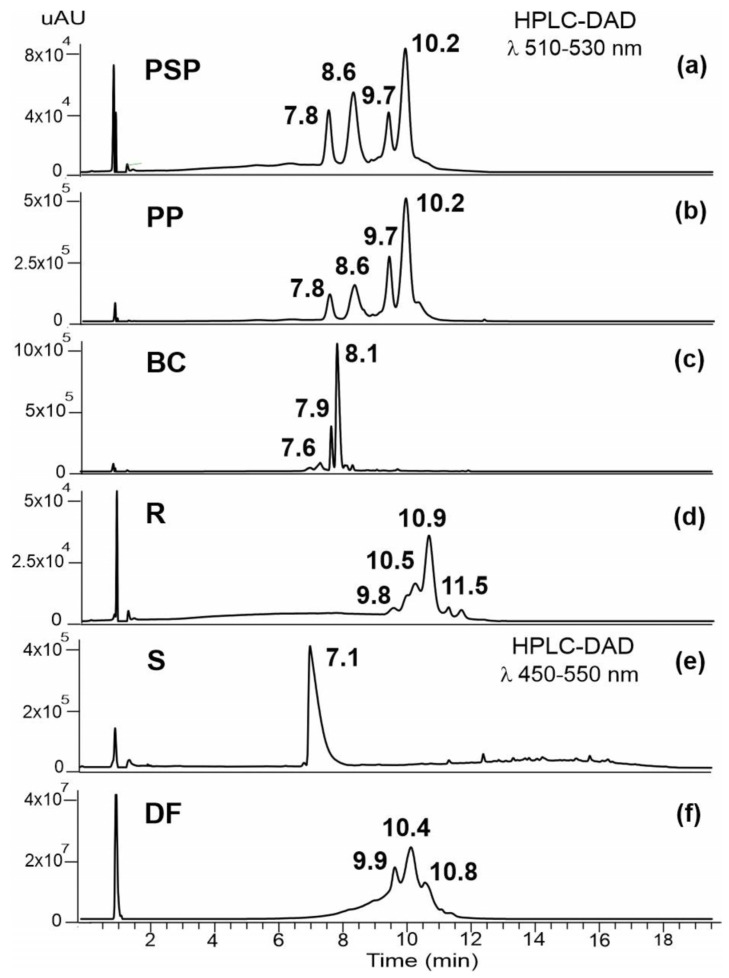
Chromatographic profile obtained between 510 and 530 nm for methanolic extracts of (**a**) purple sweet potato pulp (PSP); (**b**) purple sweet potato peel (PP); (**c**) black carrot (BC); (**d**) radish (R); and (**f**) dye factory (DF) samples. The chromatogram of the sorghum (S) sample is presented in (**e**).

**Figure 4 foods-13-02450-f004:**
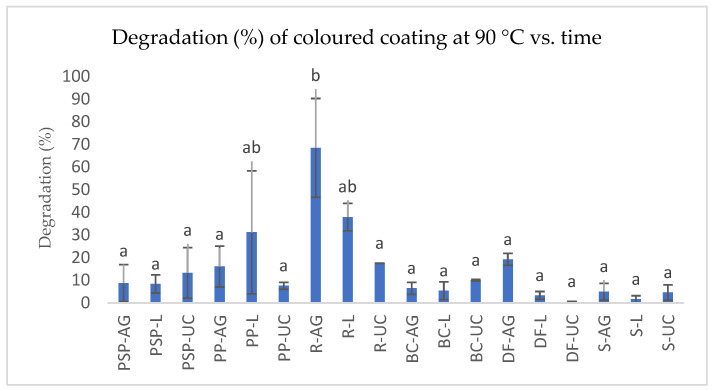
Degradation (%) of coloured coatings with extracts from purple sweet potato (PSP—Pulp; PP—Peel), black carrot (BC), radish (R), sorghum (S), and dye factory (DF), Arabic gum (AG), locust gum (L), and uncoated (UC) in sugar syrup, following temperature stability tests. Values are the mean ± standard deviation of two parallel experiments. Bars with different letters indicate significant differences (*p* < 0.05).

**Figure 5 foods-13-02450-f005:**
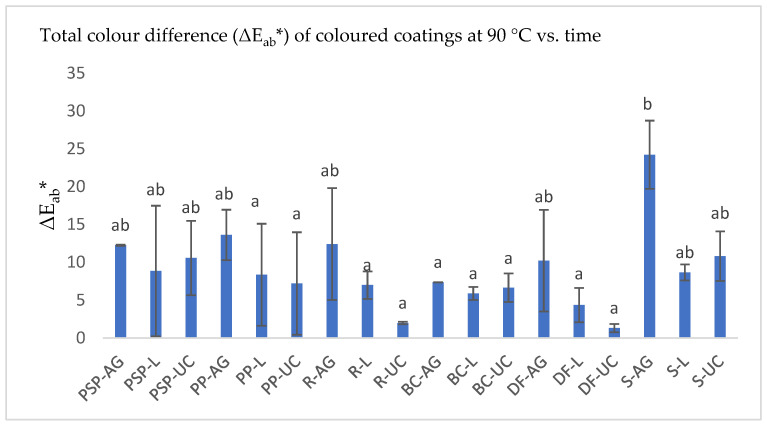
Total colour difference (ΔEab*) in coloured coatings with sweet potato peel and pulp (PP and PSP), radish peel (R), black carrot (BC), and sorghum leaf (S), incorporating Arabic gum (AG), locust gum (L), and uncoated (UC), following temperature stability tests. Values are the mean ± standard deviation of two parallel experiments. Bars with different letters indicate significant differences (*p* < 0.05).

**Figure 6 foods-13-02450-f006:**
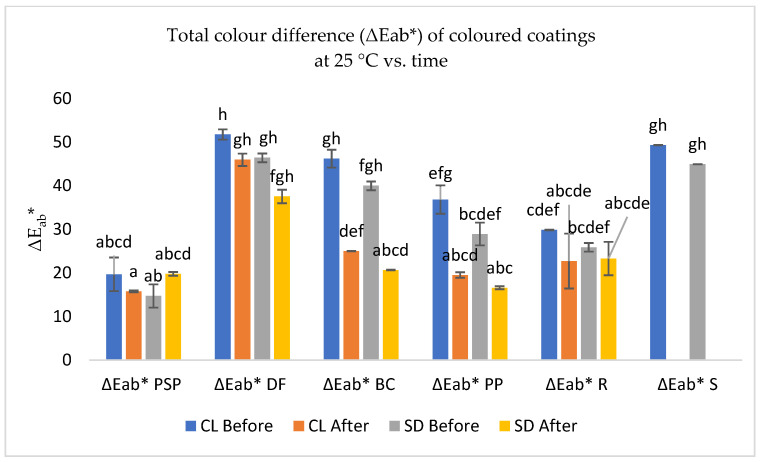
Total colour difference (ΔE_ab_*) of coloured coating extracts of purple sweet potato (PSP—pulp; PP—peel), black carrot (BC), radish (R), sorghum (S), and dye factory (DF), with almonds coloured with the clean label dye factory (DF-CL) and synthetic dyes (SDs) used as control, after 72 h of drying and 5 months of conservation. Values are the mean ± standard deviation of two parallel experiments. Bars with different letters indicate significant differences (*p* < 0.05).

**Figure 7 foods-13-02450-f007:**
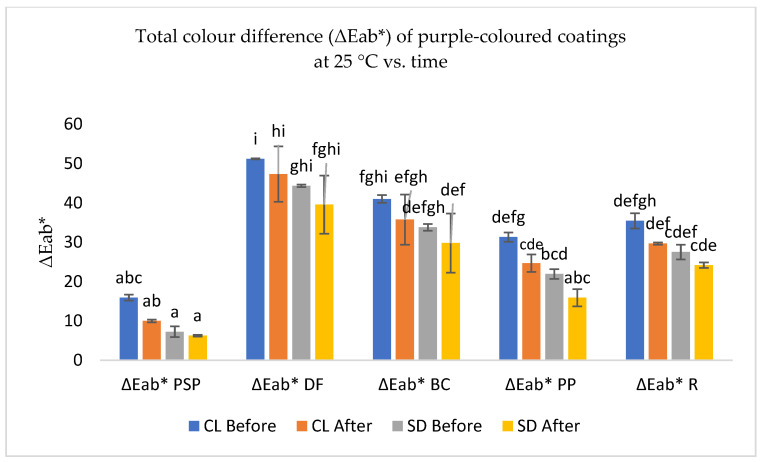
Total colour difference (ΔE_ab_*) of coloured coating extracts of purple sweet potato (PSP—Pulp; PP—Peel), black carrot (BC), radish (R), sorghum (S), and dye factory (DF), with almonds coloured with the clean label dye factory (DF-CL) and synthetic dyes (SD) used as control, after 72 h of drying and 5 months of conservation. Values are the mean ± standard deviation of two parallel experiments. Bars with different letters indicate significant differences (*p* < 0.05).

**Figure 8 foods-13-02450-f008:**
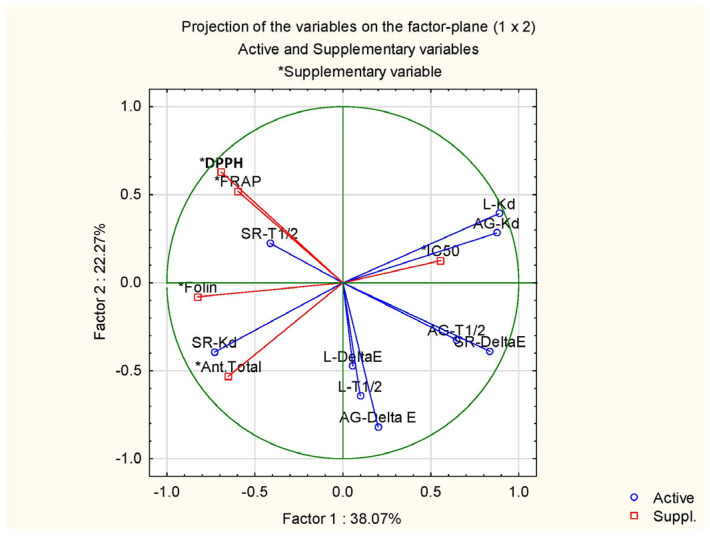
Projection of attributes on the plane.

**Figure 9 foods-13-02450-f009:**
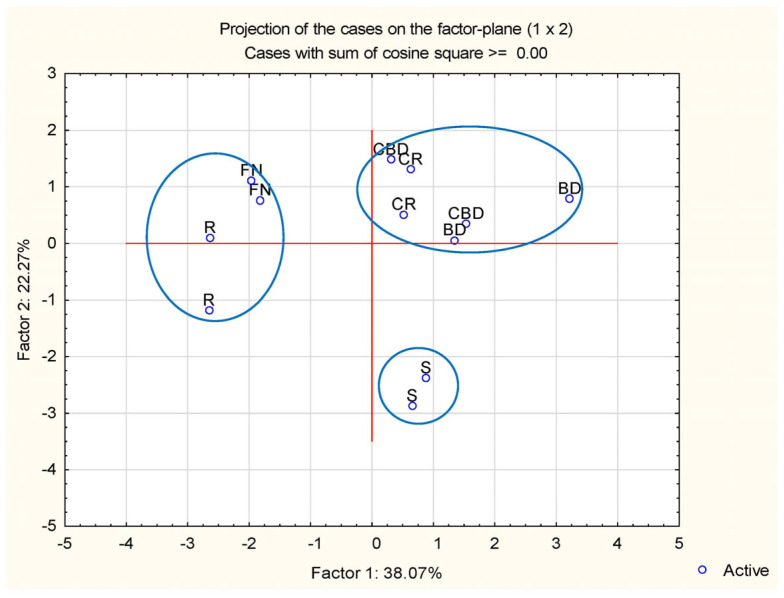
Projection of the samples on the plane defined by the two principal components. Distinct groups are marked in blue according to the type of dye.

**Table 1 foods-13-02450-t001:** Plant extracts and polymeric material in the preparation of coloured coatings.

Colouring Matter	Amount (g)	Polymeric Material	Amount (g)	Solution
Radish peel (R)	0.1	Arabic gum (AG)	0.2	sugar syrup
		Locust gum (L)		
		Uncoated (UC)		
Purple sweet potato peel (PP)	0.1	Arabic gum (AG)	0.2	sugar syrup
		Locust gum (L)		
		Uncoated (UC)		
Purple sweet potato pulp (PSP)	0.25	Arabic gum (AG)	0.2	sugar syrup
		Locust gum (L)		
		Uncoated (UC)		
Black carrot (BC)	0.05	Arabic gum (AG)	0.2	sugar syrup
		Locust gum (L)		
		Uncoated (UC)		
Sorghum (S)	0.05	Arabic gum (AG)	0.2	sugar syrup
		Locust gum (L)		
		Uncoated (UC)		
Dye factory (DF)	0.05	Arabic gum (AG)	0.2	sugar syrup
		Locust gum (L)		
		Uncoated (UC)		

**Table 2 foods-13-02450-t002:** Plant extracts and polymeric materials in the preparation of coloured coatings with blue dye addition.

Colouring Matter	Amount (g)	Polymeric Material	Amount (g)	Solution	Amount (mL)	Spirulina Blue (g)
Radish peel (R)	1.05	Locust gum	0.3	sugar syrup	50	0.34
Purple sweet potato peel (PP)	1.05	Locust gum	0.3	sugar syrup	50	0.15
Purple sweet potato pulp (PSP)	1.05	Locust gum	0.3	sugar syrup	50	0.16
Black carrot (BC)	1.05	Locust gum	0.3	sugar syrup	50	0.16
Dye factory (DF)	1.05	Locust gum	0.3	sugar syrup	50	0.25

**Table 3 foods-13-02450-t003:** Total anthocyanin content (TAC) and total phenolic compounds (TPCs) in extracts from sweet potato pulp (PSP), sweet potato peel (PP), radish peel (R), and black carrot (BC) and dye factory (DF) extracts.

Samples	TAC (mg g^−1^) DW	TPC (mg GAE g^−1^) DW
Purple Sweet Potato pulp (PSP)	1.42 ± 0.27 ^a^	6.78 ± 0.16 ^a^
Purple Sweet Potato peel (PP)	1.39 ± 0.09 ^a^	13.94 ± 1,62 ^a^
Black Carrot (BC)	0.56 ± 0.03 ^b^	31.70 ± 0.42 ^b^
Radish (R)	4.54 ± 0.10 ^c^	50.75 ± 1.36 ^c^
Sorghum (S)	0.39 ± 0.05 ^b^	39.03 ± 4.37 ^b^
Dye factory (DF)	1.72 ± 0.02 ^d^	74.53 ± 3.69 ^d^

DW: dry weight. GAE: Gallic acid equivalents. Values are the average ± standard deviation of two parallel experiments. Different letters in the same column indicate significant differences (*p* < 0.05).

**Table 4 foods-13-02450-t004:** Antioxidant and antidiabetic capacities of purple sweet potato (PSP—pulp; PP—peel), black carrot (BC), radish (R), sorghum (S), and dye factory (DF) extracts.

	DPPH^▪^	FRAP	IC_50_ (mg mL^−1^)	
Samples	(mmol TE g^−1^) DW	(mmol Fe^2^g^−1^) DW	ɑ-Amylase	ɑ-Glucosidase
Purple Sweet Potato pulp (PSP)	0.18 ± 0.006 ^c^	96.36 ± 1.91 ^e^	2.70 ± 0.08 ^c^	n.d
Purple Sweet Potato peel (PP)	0.37 ± 0.008 ^a^	246.47 ± 0.89 ^d^	2.80 ± 0.13 ^b,c^	n.d
Black Carrot (BC)	0.25 ± 0.004 ^b^	449.65 ± 0.57 ^b^	3.70 ± 0.19 ^a^	n.d
Radish (R)	0.31 ± 0.098 ^a^	309.10 ± 7.35 ^c^	3.00 ± 0.20 ^b^	n.d
Sorghum (S)	0.22 ± 0.026 ^a^	67.89 ± 2.72 ^f^	1.40 ±0.01 ^e^	0.120 ± 0.004
Dye factory (DF)	0.056 ± 0.010 ^d^	872.80 ± 13.35 ^a^	2.40 ± 0.04 ^d^	n.d.
Acarbose (positive control)	−	−	0.012 ± 0.001	0.336 ± 0.015

TE: Trolox equivalents; n.d.: not detected. Values are the average ± standard deviation of two parallel experiments. Different letters in the same column indicate significant differences (*p* < 0.05).

**Table 5 foods-13-02450-t005:** Regression equations, regression coefficient (r^2^), degradation constant (k_d_), and half-life time (t_1/2_) values for purple-coloured coatings containing extracts from sweet potato pulp (PSP), sweet potato peel (PP), radish peel (R), black carrot (BC), commercial colourant (Dye factory), sorghum (S), Arabic gum (AG), locust gum (L), and uncoated (UC) in sugar syrup, following temperature stability tests.

Extract	Equation	R^2^	K_d_	T_1/2_ (h)
PSP-AG	ln y = 9 × 10^−5^x^2^ − 0.013x + 4 × 10^−5^	R^2^ = 0.998	2.64 ± 0.90 ^d,e^	12.87 ± 1.21 ^a^
PSP-L	ln y = 0.0001x^2^ − 0.011x + 0.022	R^2^ = 0.937	1.56 ± 0.06 ^b,c,d^	8.38 ± 0.40 ^a^
PSP-UC	ln y = −0.001x − 0.005	R^2^ = 0.955	0.0025 ± 0.0021 ^a^	7.23 ± 0.61 ^a^
PP-AG	ln y = 0.0002x^2^ − 0.020x − 0.030	R^2^ = 0.959	1.78 ± 0.24 ^c,d^	4.62 ± 0.26 ^a^
PP-L	ln y = 8 × 10^−5^x^2^ − 0.0095x − 0.008	R^2^ = 0.900	1.24 ± 0.17 ^b,c^	3.23 ± 0.28 ^a^
PP-UC	ln y = −0.001x + 0.003	R^2^ = 0.949	0.0014 ± 0.0003 ^a^	8.42 ± 1.70 ^a^
R-AG	ln y = −0.016x − 0.036	R^2^ = 0.984	0.001 ± 0.004 ^a^	0.95 ± 0.25 ^a^
R-L	ln y = −0.008x + 0.012	R^2^ = 0.970	0.007 ± 0.001 ^a^	1.6 ± 0.27 ^a^
R-UC	ln y = 0.0002x^2^ − 0.025x − 0.0006	R^2^ = 0.992	3.04 ± 0.365 ^e^	3.57 ± 0.01 ^a^
BC-AG	ln y = 0.0001x^2^ − 0.011x − 0.027	R^2^ = 0.913	1.01 ± 0.61 ^a,b,c^	10.60 ± 4.37 ^a^
BC-L	ln y = 4 × 10^−5^x^2^ − 0.007x + 0.022	R^2^ = 0.951	1.20 ± 0.09 ^b,c^	15.7 ± 1.38 ^a^
BC-UC	ln y = −0.0020x + 0.006	R^2^ = 0.983	0.0019 ± 7.1 × 10^−5 a^	6.25 ± 0.23 ^a^
DF-AG	ln y = −0.0039x − 0.015	R^2^ = 0.957	0.004 ± 0.0005 ^a^	3.28 ± 0.47 ^a^
DF-L	ln y = −0.0001x^2^ + 0.006x + 0.003	R^2^ = 0.991	0.54 ± 0.004 ^a,b^	21.55 ± 1.08 ^b^
DF-UC	ln y = 1 × 10^−5^x^2^ − 0.0002x − 0.160	R^2^ = 0.996	0.52 ± 0 ^a,b^	98.83 ± 0 ^a^
S-AG	ln y = 0.0001x^2^ − 0.013x − 0.012	R^2^ = 0.986	0.71 ± 0.07 ^a,b,c^	17.47 ± 1.30 ^a^
S-L	ln y = 2 × 10^−5^x^2^ − 0.003x − 0.005	R^2^ = 0.960	0.64 ± 0.009 ^a,b^	72.83 ± 7.40 ^a,b^
S-UC	ln y = 2 × 10^−5^x^2^ − 0.003x − 0.0004	R^2^ = 0.995	0.82 ± 0.024 ^a,b,c^	18.62 ± 1.30 ^a^

Different letters in the same column indicate significant differences (*p* < 0.05).

**Table 7 foods-13-02450-t007:** Correlation coefficients between the attributes (initial variables) and the two principal components.

Variable	Principal Component 1	Principal Component 2
AG-K_d_	0.87	0.28
AG-T_1/2_	0.65	−0.32
AG-Delta E	0.20	−0.82
L-K_d_	0.89	0.39
L-T_1/2_	0.10	−0.64
L-DeltaE	0.06	−0.47
SR-K_d_	−0.73	−0.39
SR-T_1/2_	−0.41	0.22
SR-DeltaE	0.84	−0.39
*DPPH^•^	−0.69	0.63
*FRAP	−0.60	0.52
*Folin	−0.83	−0.08
*Ant.Total	−0.65	−0.53
*IC_50_	0.56	0.13

* means supplementary variables.

## Data Availability

The original contributions presented in the study are included in the article, further inquiries can be directed to the corresponding author.

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
