# Peer review of "Exploring the Potential of Anthocyanin-Based Edible Coatings in Confectionery—Temperature Stability, pH, and Biocapacity"

_foods, 2024, doi:10.3390/foods13152450_

Round 1

Reviewer 1 Report

Comments and Suggestions for Authors

The article is not of sufficient quality. Specific questions and points requiring attention are itemized below.

1. Line 76. Describe the polymers more used in the fabrication of coating materials. 

2. It still needs to be determined the novelty of the work compared to what has already been published. What is the difference between what is published and what the authors want to publish? It is not clear. This information must be added in the introduction section. 

3. Line 152. What liquid was used to extract?? Water?? Ethanol?? Please describe.

4. Line 150. The authors must carry out a bromatological analysis of the extract obtained. 

5. Line 172. Vacuum conditions of the freeze-drying process.

6. In Table 1, the authors describe the plant extracts and polymeric material in preparing colored coatings. However, Polymeric materials such as arabic gum and Locust gum are not defined and justified in the introduction section.

7. Why do the authors use Arabic gum and Locust gum as polymeric materials to fabricate the coating materials?? 

8. Could you please elaborate on the reasons for choosing sugar syrup for the coating's fabrication? This will help the readers understand the rationale and be convinced of its suitability. 

9. Line 384. What was the bibliographic reference used? Has anyone else done it?

10. Line 435. Remember, when a Reference is used under the text, the name or surname must be described, while when it is cited at the end of the paragraph, only the reference number is added.

11. The discussion section is poor. I think more comparisons with previous literature should be discussed.

12. Table 4 can be put in the supplementary material.

Author Response

Thank you very much for taking the time to review our manuscript titled “Exploring the Potential of Anthocyanin-Based Edible Coatings in Confectionery – Temperature Stability, pH and Biocapacity”. We truly appreciate the effort and expertise you have contributed. The answers to the questions are as follows:

  1. Line 76. Describe the polymers more used in the fabrication of coating materials.

Answer: The polymers more used in the fabrication of coating materials were described according the suggestion.

  1. It still needs to be determined the novelty of the work compared to what has already been published. What is the difference between what is published and what the authors want to publish? It is not clear. This information must be added in the introduction section.

Answer: Thank you for your valuable suggestions.

The innovation in this study is to obtain a stable purple colour directly from anthocyanin-based plant matrices, specifically using leaves and peels from agri-food by-products. Additionally, we are investigating the stabilization of anthocyanin-based dyes using gum Arabic and locust bean gum coatings. Another objective is to evaluate the resilience of these coatings to temperature variations in terms of colour discoloration when applied to sugar syrup used in the confectionery industry.

This information has been added to the introduction section according to the reviewer suggestions.

  1. Line 152. What liquid was used to extract?? Water?? Ethanol?? Please describe.

Answer: The liquid used for extraction is water as described in section 2.3

  1. Line 150. The authors must carry out a bromatological analysis of the extract obtained.

Answer: Thank you for your valuable feedback. We understand the importance of a comprehensive bromatological analysis, which typically involves comprehensive nutritional and composition profiling, in certain contexts. However, our research is centred on colour stability and the effectiveness of natural gums in preserving this colour under varying temperature conditions. Furthermore, the percentage of the extract that is incorporated into the final product is very low, so its contribution to the nutritional value of the almonds is not relevant. Given these objectives, our work does not require a detailed bromatological analysis.

  1. Line 172. Vacuum conditions of the freeze-drying process.

Answer: The vacuum pressure of the freeze drier was set at 0.2 hPa, the plate temperature was 20 °C, and the condenser was at –50 °C for 24 h

  1. In Table 1, the authors describe the plant extracts and polymeric material in preparing coloured coatings. However, Polymeric materials such as Arabic gum and Locust gum are not defined and justified in the introduction section.

Answer: The Polymeric materials such as Arabic gum and Locust gum were defined and justified in the introduction section, according the suggestion.

  1. Why do the authors use Arabic gum and Locust gum as polymeric materials to fabricate the coating materials??

Answer: The use of Arabic gum and Locust gum as polymeric materials to fabricate the coating materials is justified by the ability to form gels, temperature resistance, solubility in water and has antioxidant capacity, as is explained in the introduction section.

  1. Could you please elaborate on the reasons for choosing sugar syrup for the coating's fabrication? This will help the readers understand the rationale and be convinced of its suitability.

Answer: Sugar syrup was used in the factory manufacturing process to cover the almonds and was adapted in the study for comparison. Sugar acts as a humectant, retaining moisture and preventing the product from drying out, prolonging the shelf life of the almonds. Furthermore, high concentrations of sugar inhibit microbial growth, preserving the product for longer periods. It also serves as a binding agent with anthocyanin extracts, ensuring that the coating material adheres to the substrate facilitating handling and storage.

  1. Line 384. What was the bibliographic reference used? Has anyone else done it?

Answer: The methodology for the formulation of polymeric coatings was adapted from the factory manufacturing process.

  1. Line 435. Remember, when a Reference is used under the text, the name or surname must be described, while when it is cited at the end of the paragraph, only the reference number is added.

Answer: In the manuscript we use the journal's rules. This way, only numbers are allowed.

  1. The discussion section is poor. I think more comparisons with previous literature should be discussed.

Answer: The discussion was revised according to the suggestions.

  1. Table 4 can be put in the supplementary material.

Answer: The table was removed according the suggestion

Reviewer 2 Report

Comments and Suggestions for Authors

Dear Authors,

The manuscript is well written, defined and clear in terms of objectives. Some details, typing, need to be corrected and some other doubts are present in the attached file. I suggest reviewing the conclusions. Initially they are very extensive, they can be reduced and more objective. The results are already presented and discussed, without the need for redundancy. Some paragraphs are not objectives of the work and are not part of the conclusions, but rather as a conclusion to the discussion section. I hope I have contributed to improving the manuscript and the reader.

Author Response

Thank you very much for taking the time to review our manuscript titled “Exploring the Potential of Anthocyanin-Based Edible Coatings in Confectionery – Temperature Stability, pH and Biocapacity”. We truly appreciate the effort and expertise you have contributed. The answers to the questions are as follows:

 Line 166: In the maceration method, the extraction.

Answer: The sentence was corrected as suggested: “For the solvent extraction, the mixture…”

Line 171: under vacum?

Answer: The sentence was corrected as suggested: at reduced pressure

Line 307: According to the instructions for authors, units should be converted to International System of Units (SI): mcgmL-1

Answer: According to SI, the symbol for microgram is μg.

Line 341: 3DXAs

Answer: The abbreviation was corrected to 3-DXAs as suggested.

Line 357: 1h15 min

Answer: The abbreviation was corrected to 75 min, and not 95 min as suggested.

Line 386: 3DXAs

Answer: The abbreviation was corrected to 3-DXAs as suggested.

Line 388: 5cmx5cm

Answer: The units was corrected to 5 cm2 as suggested.

Line 389: minute

Answer: The word was corrected to min as suggested.

Line 390: the °C

Answer: The units was removed as suggested.

Line 422: Remove the units nm.

Answer: The units were removed as suggested.

Line 427: The abbreviation is incorrect

Answer: The abbreviation was corrected as suggested.

Line 432: Remove the point.

Answer: The point was removed as suggested.

Line 454: The abbreviation is incorrect

Answer: The abbreviation was corrected as suggested.

Line 484: The reference must be according the journal rules.

Answer: The reference was corrected according the journal rules.

Line 527: May be 36-41?

Answer: The references were corrected.

Line 537: May be 42-45?

Answer: The references were corrected.

Line 547: May be 46-48?

Answer: The references were corrected.

Line 626: Activities or capacities?

Answer: The word was corrected as suggested.

Line 626: Activity or capacity?

Answer: The word was corrected as suggested.

Line 628: Activity or capacity?

Answer: The word was corrected as suggested.

Line 635: The abbreviation is incorrect

Answer: The abbreviation was corrected as suggested.

Line 640: Activity or capacity?

Answer: The word was corrected as suggested.

Line 647: The abbreviation is incorrect

Answer: The abbreviation was corrected as suggested.

Line 651: Activity or capacity?

Answer: The word was corrected as suggested.

Line 652: May be 52-55?

Answer: The references were corrected.

Line 659: May be 57-58?

Answer: The references were corrected.

Line 663: activity?

Answer: The word was corrected as suggested.

Line 664: activity?

Answer: The word was corrected as suggested.

Line 666: activity?

Answer: The word was corrected as suggested.

Line 669: activity?

Answer: The word was corrected as suggested.

Line 671: activity?

Answer: The word was corrected as suggested

Line 664: activity?

Answer: The word was corrected as suggested.

Line 672: activities?

Answer: The word was corrected as suggested.

Line 675: activity?

Answer: The word was corrected as suggested.

Line 690: activity?

Answer: The word was corrected as suggested.

Line 712: italic?

Answer: The word was corrected.

Line 714: hours

Answer: The word was corrected.

Line 718: hours?

Answer: The word was corrected.

Line 883: Activity or capacity?

Answer: The word was corrected as suggested.

Line 884: Activity or capacity?

Answer: The word was corrected as suggested.

Line 891-919: Very extensive!

Answer: The sentence was reformulated as suggested.

Line 915: 1h15minutes

Answer: The word was corrected to 75 min.

Line 919: hours

Answer: The word was corrected.

Line 924-926: send to the end of the section

Answer: The paragraph was moved as suggested.

Reviewer 3 Report

Comments and Suggestions for Authors

The authors have chosen an interesting and current topic, the development of purple-coloured polymeric coatings using natural anthocyanin and desoxyanthocianidins (3-DXA) colourants for application to chocolate almonds.

But the work is not sown in some places, it needs to be either reorganized and clarified.

Check the sentence in lines30-31.

In the work, the authors used chocolate almonds as a product. However, in part 2.8, coatings were applied onto white solid sugar pastes, each cut into 5cmx5cm squares. Why?

Further, authors wrote in section 2.6, that "control was a commercial colourant (Dye factory) of unknown composition." Still, in part 2.9 authors compared coting pastes (why pastes and not almonds?) with purple Easter almonds dyed with clean label colourants and those with synthetic dyes.

Table 4 is too extensive. Can it be presented differently or given as supplementary?

It is necessary to match the order of the abbreviations in the name of table 6 with the order of the examples shown in the table.

It is not clear what the R-UC to DF-US samples are. If they are uncoated, where are the anthocyanins? In sugar syrup and how were they applied? As a coating? Are these the results of the kinetics degradation of the colored coatings, without the product? What's the point of that experiment?

Abbreviations for S and DF are missing in the name of table 6.

Author Response

Thank you very much for taking the time to review our manuscript titled “Exploring the Potential of Anthocyanin-Based Edible Coatings in Confectionery – Temperature Stability, pH and Biocapacity”. We truly appreciate the effort and expertise you have contributed. The answers to the questions are as follows:

 Check the sentence in lines 30-31.

“Maceration and ultrasounds (US) were used at different pH were used to 30 extract anthocyanidins and 3-DXA from the plant material.”

Answer: The sentence was replaced as suggested: “Anthocyanidins and 3-DXA were extracted from the plant material using solvent extraction and ultrasound-assisted (UA) methods at different pH values”.

In the work, the authors used chocolate almonds as a product. However, in part 2.8, coatings were applied onto white solid sugar pastes, each cut into 5cmx5cm squares. Why?

Answer: The chocolate almonds, provided by the factory, underwent a coating process involving grinding. This process requires a substantial amount of dye to color the almonds. Therefore, we decided to apply the same dyes, supplied by the factory, to white solid sugar pastes for comparison.

Further, authors wrote in section 2.6, that "control was a commercial colourant (Dye factory) of unknown composition." Still, in part 2.9 authors compared coting pastes (why pastes and not almonds?) with purple Easter almonds dyed with clean label colourants and those with synthetic dyes.

Answer: The chocolate almonds supplied by the factory were coloured with two types of dyes: a commercial dye (Dye factory) of unknown composition called Clean Label and a synthetic dye. The commercial dye was also applied to sugar pastes to compare with dye extracts from the matrices under study. Thus, the commercial colouring served as a control for both the chocolate almonds and the sugar pastes.

 Table 4 is too extensive. Can it be presented differently or given as supplementary?

Answer: The Table 4 will be put in supplementary.

It is necessary to match the order of the abbreviations in the name of table 6 with the order of the examples shown in the table

Answer: Corrections were made as suggested.

It is not clear what the R-UC to DF-US samples are. If they are uncoated, where are the anthocyanins? In sugar syrup and how were they applied? As a coating? Are these the results of the kinetics degradation of the colored coatings, without the product? What's the point of that experiment?

Answer: In this experiment, the stability of anthocyanin extracts at 90 ºC over time was studied Extracts without polymeric coatings were compared to those coated with gum Arabic and locust gum. This comparison aimed to determine which coating provided better protection for the anthocyanins and resulted in less color variation. Without the uncoated extracts, such a comparison would not have been possible.

Abbreviations for S and DF are missing in the name of table 6.

Answer: Corrections were made as suggested.

Reviewer 4 Report

Comments and Suggestions for Authors

The aim of the study was to compare effect of the potential of anthocyanin-based edible coatings in confectionery. The objectives and background were well stated. However, there are a few comments that need to be addressed prior to the publication.

1. In the abstract, the content, methods and results of this experimental study should be described more briefly and accurately, and the experimental results should also be appropriately described in the abstract

2. In the keywords, I suggest 3-5.

3. In the introduction, I suggest paragraphs 3-5 instead of 12.

4. Some paragraphs have first line indent and some paragraphs do not have first line indent, please comb the full text and unify.

5. In Materials and Methods, please supplement the production process of each sample.

6. I suggest the conclusion should be concise.

Comments on the Quality of English Language

None.

Author Response

Thank you very much for taking the time to review our manuscript titled “Exploring the Potential of Anthocyanin-Based Edible Coatings in Confectionery – Temperature Stability, pH and Biocapacity”. We truly appreciate the effort and expertise you have contributed. The answers to the questions are as follows:

  1. In the abstract, the content, methods and results of this experimental study should be described more briefly and accurately, and the experimental results should also be appropriately described in the abstract.

Answer: The abstract was modified as suggested.

  1. In the keywords, I suggest 3-5.

Answer: The keywords were reduced as suggested.

  1. In the introduction, I suggest paragraphs 3-5 instead of 12.

Answer: The paragraph 12 was removed as suggested.

  1. Some paragraphs have first line indent and some paragraphs do not have first line indent, please comb the full text and unify.

Answer: The paragraphs were uniformized according the suggestions.

  1. In Materials and Methods, please supplement the production process of each sample.

Answer: The production process for each sample is complemented with flowcharts as requested.

  1. I suggest the conclusion should be concise.

Answer: The conclusion was modified according suggestions.

It is necessary to match the order of the abbreviations in the name of table 6 with the order of the examples shown in the table

Answer: Corrections were made as suggested

Round 2

Reviewer 1 Report

Comments and Suggestions for Authors

Accepted 

Reviewer 3 Report

Comments and Suggestions for Authors

Thanks to the authors for their efforts to provide clarifications and correct the work according to suggestions.